# CoPruning: Exploring the Parameter-Gradient Nonlinear Correlation for Neural Network Pruning Using Copula Function

## Abstract

The sheer size of modern neural networks necessitates pruning techniques to overcome the significant computational challenges posed by model serving. However, existing pruning techniques fail to capture the nonlinear correlation between parameters and gradient, which is crucial in the pruning process, thus leading to low accuracy under high sparsity. In this work, we propose CoPruning, a new pruning framework, which uses a copula function based joint distribution model that precisely captures the intricate nonlinear correlation between parameters and gradient, enabling more insightful pruning decisions. Additionally, we integrate a local optimization approach within CoPruning to better capture relative change in parameters within their local context, providing new metrics for achieving finer-grained optimization. Extensive experiments on various networks reveal CoPruning's comparable performance to state-of-the-art (SoTA) pruning algorithms. CoPruning outperforms the SoTA with **3.09%**, **1.87%**, and **2.19%** higher accuracy on MLPNet, ResNet20, and ResNet50 at 0.98 sparsity, respectively, and **10.43%** higher accuracy on MobileNetV1 at 0.9 sparsity on ImageNet.

## 1 INTRODUCTION

Neural networks have emerged as a cornerstone of modern machine learning, facilitating groundbreaking advancements across diverse application domains, spanning from image recognition to natural language processing (Devlin et al., 2019; OpenAI, 2023). Nevertheless, as networks become larger and more intricate, e.g., ChatGPT, they impose substantial computational burdens, presenting formidable challenges in deployment, especially in resource-constrained environments such as mobile devices and embedded systems (Chen et al., 2016). To tackle these challenges, network pruning is regarded as a key technique to reduce the size and storage requirements of neural networks while maintaining or minimally impacting their performance by removing redundant parameters (such as neurons, filters, or connections) from the network (Li et al., 2017). In the field of neural network pruning, pruning methods are generally categorized into structured pruning and unstructured pruning. **Structured pruning** typically retains the original structure of the neural network while reducing unnecessary connections and weights, improving computational efficiency without significantly degrading performance. For instance, Huang & Lee (2022) proposed a strategy for training structured neural networks through manifold identification and variance reduction, while Molchanov et al. (2017c) focused on pruning convolutional neural networks (CNNs) for resource-efficient inference. Other relevant works include Sze et al. (2017) and Anwar et al. (2017), who further explore structured pruning in neural networks. Additionally, Fang et al. (2023) and Yang et al. (2024) studied structured pruning in large-scale language models, while He et al. (2017) focused on accelerating very deep neural networks through channel pruning. On the other hand, **unstructured pruning** allows the removal of individual weights, often resulting in a more sparse network. For example, He et al. (2022) investigated how sparse pruning exacerbates overfitting, while Han et al. (2016) studied deep compression and its efficient inference engine. Similarly, Molchanov et al. (2017a) utilized variational dropout to sparsify deep neural networks, and Frantar et al. (2022) ensured speedup guarantees through accurate pruning. Other notable unstructured pruning methods include Guo et al. (2016), who proposed dynamic network surgery, and Aghasi et al. (2017), who introduced convex pruning methods that provide performance guarantees for deep neural networks.

In neural networks, the relationship between model parameters and gradient is typically nonlinear, influenced by several factors. First, the use of *nonlinear activation functions*, such as ReLU and Sigmoid, results in a nonlinear mapping between the network's input and output, which directly impacts the correlation between the parameters and gradient Zubair & Singha (2020a). Second, the *backpropagation algorithm* propagates gradient through multiple layers, where each layer's nonlinear transformation further enforces this nonlinear relationship (Agarwal & Ramampiaro, 2024a). Moreover, the *loss function* is often nonlinear (e.g., cross-entropy or mean square error), meaning that the gradient's relationship with parameters is highly dependent on the nonlinearity of the loss itself Zubair & Singha (2020b). Finally, *optimization algorithms*, such as Adam and RMSprop, dynamically adjust learning rates based on the gradient's history, further complicating and nonlinearizing the parameter updates (Agarwal & Ramampiaro, 2024b). These factors collectively contribute to the nonlinear correlation between parameters and gradient in deep learning models.

Applying nonlinear relationships to predict models can effectively enhance the performance of models (Kulathunga et al., 2020). Many traditional pruning methods rely on relatively simple criteria, such as magnitude-based thresholds (Han et al., 2015) or heuristic techniques (Molchanov et al., 2017b). While these approaches have proven effective in various scenarios, they often simplify the modeling of dependencies between network parameters and gradient, which may not adequately account for the complex, nonlinear correlations present in neural networks. As a result, this simplification can lead to less accuracy.

We propose `CoPruning`, which is a novel framework that use copula function to model the nonlinear correlation between parameters and gradient. By disentangling the marginal distributions from their dependency structure, copulas offer a more flexible and accurate means of capturing nonlinear interactions. This challenges the traditional linear assumptions and provides a more precise foundation for pruning decisions. `CoPruning` integrates a local optimization approach to capture relative changes in parameters within their local context. Through rigorous experiments, we show that `CoPruning` outperforms traditional pruning methods across multiple performance metrics. Our method achieves not only higher pruning efficiency but also significantly improves model accuracy and robustness, particularly under extreme sparsity, highlighting its superiority in practical applications.

## 2 RELATED WORK AND PROBLEM SETUP

### 2.1 RELATED WORK

Neural network pruning is an essential technique for reducing the complexity of deep neural networks, thereby facilitating their seamless deployment in resource-constrained environments. The key to effective pruning lies in assessing the impact on the loss function $E$ when specific weights are removed, typically quantified by $\Delta E$. A lot of methods have been developed, ranging from straightforward first-order approaches to more sophisticated second-order methods that utilize the Hessian matrix to guide pruning decisions. One of the earliest methods to introduce Hessian-based pruning is proposed by LeCun et al. (1989) with the Optimal Brain Damage (OBD) method. OBD uses a second-order derivative (the diagonal elements of the Hessian matrix) to estimate the impact on the loss function $\Delta E$ when a weight, denoted by $w_q$, is removed. The formulation of $\Delta E$ is expressed as

$$\Delta E \approx \frac{1}{2} h_{qq} w_q^2, \tag{1}$$

where $h_{qq}$ represents the diagonal elements of the Hessian matrix $\mathbf{H}$, which is the second derivative of the loss function $E$ with respect to the weight $w_q$. However, OBD assumes that the Hessian matrix is diagonally dominant, and ignores the interdependency between weights. To address this limitation, Hassibi & Stork (1992) introduced the Optimal Brain Surgeon (OBS) method. OBS expands on OBD by taking into account the full Hessian matrix instead of just the diagonal elements. In addition, OBS considers the interaction between different weights, thus leading to a more precise estimation of the pruning impact on the network's performance. With OBS, $\Delta E$ is derived by employing the inverse of the Hessian matrix, which is expressed as

$$\Delta E \approx \frac{w_q^2}{2[H^{-1}]_{qq}}. \tag{2}$$

The Combinatorial Brain Surgeon (CBS) method, introduced by Yu et al. (2022), proposes a combinatorial optimization approach that considers the interaction between multiple weights. The optimization problem for CBS is formulated as

$$\min \frac{1}{2} \sum_{i=1}^{N} \sum_{j=1}^{N} (w_i - \bar{w}_i) H_{ij} (w_j - \bar{w}_j). \tag{3}$$

In equation (3), $\bar{w}_i$ represents the weights before pruning, and $H_{ij}$ are the elements of the Hessian matrix that capture the interaction between weights $w_i$ and $w_j$. The indices $i$ and $j$ represent different weights in the neural network.

The CHITA method, proposed by Benbaki et al. (2023), efficiently approximates the Hessian matrix as

$$H \approx \frac{1}{n} \sum_{i=1}^{n} \nabla \ell_i \nabla \ell_i^T = \frac{1}{n} G^T G. \tag{4}$$

Here, $G = [\nabla \ell_1, \ldots, \nabla \ell_n]^T$ is the matrix containing the gradients for each sample, and $\ell_i$ represents the loss for the $i$-th sample. This method significantly reduces computational complexity and is suitable for large-scale network pruning due to avoiding explicit computation of the Hessian matrix.

Chen et al. (2022) formulates the pruning problem as a ridge regression task. The formulation is expressed as

$$\min_{w} Q(w) = \frac{1}{2} \|y - Xw\|^2 + \frac{n\lambda}{2} \|w - \bar{w}\|^2. \tag{5}$$

In (5), $X$ is the gradient matrix, $y$ is the product of the reference weights $\bar{w}$ and $X$, and $\lambda$ is the regularization parameter. This encourages the weights $w$ to stay close to the reference weights $\bar{w}$, which helps to prevent overfitting and enhance generalization.

Based on the framework of CHITA and ridge regression, You & Cheng (2024) proposes a SWAP method that combines linear regression (LR) with optimal transport principles to address noisy pruning scenarios. LR integrates the techniques from Benbaki et al. (2023) and Chen et al. (2022) and is expressed as:

$$\min_{w} Q(w) = \frac{1}{n} \sum_{i=1}^{n} \|x_i(w) - y_i\|^2 + \lambda \|w - \bar{w}\|^2 \tag{6}$$

where $x_i(w) = w^T \nabla \ell_i$ represents the modeling result of the model correlation given the current weights $w$, and $y_i = \bar{w}^T \nabla \ell_i$ is the target output, with $\bar{w}$ being the reference weights.

For noisy pruning scenarios, You & Cheng (2024) developed Entropic Wasserstein Regression (EWR), which enhances the robustness of pruning by integrating optimal transport with entropy regularization. The complete optimization problem for EWR is formulated as:

$$\min_{w} Q(w) = \inf_{\Pi \in \Pi} \left\{ \sum_{i=1}^{n} \sum_{j=1}^{n} \|x_i(w) - y_j\|^2 \pi_{ij} + \epsilon \sum_{i=1}^{n} \sum_{j=1}^{n} \log \left( \frac{a_i b_j}{\pi_{ij}} \right) \pi_{ij} \right\} + \lambda \|w - \bar{w}\|^2 \tag{7}$$

Here, $\pi_{ij}$ represents the transport plan, describing how mass is transferred from $x_i(w)$ to $y_j$, minimizing transport cost. The parameters $a_i$ and $b_j$ correspond to marginal distributions in the optimal transport problem, and $\epsilon$ controls the strength of the entropy regularization term, ensuring smoother solutions.

This approach provides a more robust solution to pruning in noisy environments by minimizing the impact of noise while maintaining performance.

## 2.2 PROBLEM SETUP

To enhance the effectiveness of network pruning, particularly in capturing complex dependencies between parameters, we propose `CoPruning`. Our approach introduces a Frank Copula-based joint distribution and incorporates a local neighborhood analysis of the parameter vector $\mathbf{w}$.

In the modified objective function (8), $C_i(\mathbf{s}, k_r)$ represents the Frank Copula function, capturing the joint distribution between the localized gradient results $s$ and the localized weight sum $k_r$. The term

$\mathcal{H}_c(\mathbf{s}, k_r)$ represents the Copula entropy, measuring the difference in entropy between the current and reference distributions. Here, $s$ stands for the localized gradient results, reflecting the influence of localized gradient changes in the parameter space, and $k_r$ represents the localized weight sum ratio, which is derived from the weight vector $\mathbf{w}$ within a specific neighborhood radius. The symbol $\bar{k}_r$ is the reference localized weight sum, and $\mathbf{w}$ is the parameter vector before optimized. These components together form the core of our Copula-based pruning framework.

$$\min_{k} Q(k) = \sum_{i=1}^{n} \|C_i(\mathbf{s}, k_r) - C_i(\mathbf{s}, \bar{k}_r)\|^2 + \alpha \|\mathcal{H}_c(\mathbf{s}, k_r) - \mathcal{H}_c(\mathbf{s}, \bar{k}_r)\|^2 + \lambda \|k_r - \bar{k}_r\|^2 \quad (8a)$$

$$\text{s.t.} \quad 0 \leq s, k, k_r \leq 1 \quad (8b)$$

## 3 COPULA FUNCTION BASED PRUNING FRAMEWORK

In this section, we present the theoretical foundation of CoPruning. To enable the fitting of the Copula function, we must process the gradient and parameters to meet specific requirements. In particular, the Frank Copula necessitates that the variables $u$ and $v$ are constrained within the interval [0,1]. This normalization step transforms the gradient and parameters to lie within this range, aligning with the requirements for copula fitting. Next, we introduce Sklar's Theorem, which provides the mathematical basis for applying Copula functions in modeling dependencies. Subsequently, we explain how the Frank Copula and Copula entropy are integrated into our pruning method to manage both local and global dependencies in the network.

In our optimization objective, we define $K_{\text{I}}$ and $K_{\text{II}}$ for (8) as follows:

$$K_{\text{I}} = \sum_{i=1}^{n} \|C_i(\mathbf{s}, k_r) - C_i(\mathbf{s}, \bar{k}_r)\|^2 \quad (9)$$

where $C_i$ represents the Copula function, which is employed to handle the joint distribution of the localized parameters $k_r$ and $\bar{k}_r$ along with the localized gradients $s$.

$$K_{\text{II}} = \alpha \|\mathcal{H}_c(\mathbf{s}, k_r) - \mathcal{H}_c(\mathbf{s}, \bar{k}_r)\|^2 \quad (10)$$

where $H_c$ represents the Copula entropy function, which representing the amount of information contained in the constructed joint distribution of the localized parameters $k_r$ and $\bar{k}_r$ along with the localized gradients $s$.

### 3.1 LOCAL OPTIMIZATION FOR PRUNING

To enhance the efficiency of the pruning process, we introduce Local optimization, a technique that allows us to focus on specific regions within the parameter matrix, thereby enabling a more efficient capture of local dependencies. This approach also facilitates the subsequent handling of parameters and gradient for Copula fitting. In equation $Q(k)$, the localized model parameters are represented as $k_r$, while the pre-pruning parameters are denoted as $\bar{k}_r$. Additionally, the gradient information is represented by $\mathbf{s}$.

Using a sliding window of size $r \times r$, we normalize the parameter values based on their local neighborhoods. For each element $W_{i,j}$ in the parameter matrix, we compute the sum of the absolute values within the window centered at $W_{i,j}$:

$$Sum_{i,j} = \sum_{m=-\lfloor r/2 \rfloor}^{\lfloor r/2 \rfloor} \sum_{n=-\lfloor r/2 \rfloor}^{\lfloor r/2 \rfloor} |W_{i+m,j+n}|. \quad (11)$$

Each element is then updated according to the following formula:

$$W'_{i,j} = \frac{|W_{i,j}|}{Sum_{i,j}}. \quad (12)$$

Figure 1: The Local optimization process applied to a parameter matrix.

This normalization not only ensures that the gradient and parameters fall within the required interval [0,1], but also highlights the relative importance of parameters within localized regions. This sensitivity to local dependencies allows the pruning process to remain both efficient and effective. As illustrated in Figure 1, the parameter $r$ denotes the radius of the local window used for calculating the localized weight sum, which captures the dependencies within a specific neighborhood of the parameter space.

### 3.2 SKLAR'S THEOREM AND COPULA FUNCTIONS

Copula function is fundamentally grounded in Sklar's Theorem, which provides a framework for understanding the correlations between multivariate distributions. By separating the marginal distributions from their dependence structure, Copula functions enable the analysis of complex dependencies among random variables while preserving the individual characteristics of each marginal distribution.

**Proposition 1 (Sklar's Theorem)** Sklar's Theorem (Sklar, 1959) states that any multivariate joint distribution can be decomposed into its marginal distribution function and a Copula function. Specifically, denote by $H(x_1, x_2, \ldots, x_n)$ an $n$-dimensional joint distribution function, and denote by $F_1(x_1), F_2(x_2), \ldots, F_n(x_n)$ its marginal distribution functions. Then there exists a $n$-dimensional Copula function $C$ such that for all $x_1, x_2, \ldots, x_n$,

$$H(x_1, x_2, \ldots, x_n) = C(F_1(x_1), F_2(x_2), \ldots, F_n(x_n))$$

Here, the Copula function $C$ captures the dependency structure between the marginal distributions. This allows for the separate study of marginal distributions and the correlation structure between variables, thus providing a powerful tool for analyzing complex multivariate distributions. This characteristic makes Copula functions exceptionally valuable in our framework, as preserving the dependency structure between network parameters is crucial for maintaining performance during pruning.

Next, I will introduce the copula function used in the pruning process. Copula functions can be classified into various types, such as Gaussian, t, and Archimedean copulas. Each type is suited for different dependency structures among variables. Among these, we utilize the Frank Copula, which is particularly advantageous due to its ability to model both positive and negative dependencies without imposing strict restrictions on the marginal distributions. This flexibility is crucial in our optimization framework, where the strength of dependencies can vary across different regions of the network.

We use the Frank Copula, a type of Archimedean copula characterized by the parameter $\theta$. A key property of the Frank Copula is that as $\theta$ approaches zero, it degenerates into the independence copula, which means the variables become independent. This characteristic is crucial in our optimization framework, where the strength of dependencies can vary across different regions of the

network. The Frank Copula function in (9) is expressed as

$$C_\theta(u, v) = -\frac{1}{\theta} \log \left( 1 + \frac{(e^{-\theta u} - 1)(e^{-\theta v} - 1)}{e^{-\theta} - 1} \right) \tag{13}$$

where $u$ and $v$ are the marginal distributions of the variables, and $\theta$ controls the strength of the dependence. To estimate $\theta$, we use Kendall's tau ($\tau_b$), which measures the correlation between two variables. The relationship between Kendall's tau and $\theta$ is expressed as

$$\tau_b = 1 - \frac{4}{\theta} \left( 1 - \frac{1}{\theta} \int_0^\theta \frac{t}{e^t - 1} \, dt \right) \tag{14}$$

Given Kendall's tau, we can estimate $\theta$ numerically.

### 3.3 CORRELATION MODELING USING COPULA

The Copula function can capture the nonlinearity, asymmetry, and correlation of distribution tails between variables, which plays an important role in extreme value analysis and predicting extreme events (Nelsen, 2000). A simple proof of the nonlinear correlation between parameters and gradients is provided in A.1.

Next, we will explain the process of modeling dependence using copulas. The analysis of modeling nonlinear dependence using copula functions is primarily featured in Karra (2018). Moreover, The correlation between copulas and tail dependence coefficients is essential for understanding how extreme events are correlated between two variables. Copula functions describe the joint dependence structure between two random variables, separating the marginal distributions from the dependency structure. In cases where we are particularly interested in the behavior of variables in the tails of their distributions, tail dependence coefficients provide a quantitative measure of how likely it is that both variables take on extreme values simultaneously. These coefficients can be computed based on the copula function, which allows us to examine the upper and lower extremes of joint distributions.

The upper tail dependence coefficient ($\lambda_U$) describes the degree of dependence between two variables in the upper tails of their distributions, where both variables take on large values. In terms of copulas, this coefficient can be expressed as:

$$\lambda_U = \lim_{w \to 1^-} \frac{1 - 2w + C(w, w)}{1 - w}$$

Here, $w$ represents the probability level as both marginal distributions approach their upper extremes (i.e., $u, v \to 1$), and $C(w, w)$ is the copula function that captures the joint distribution of the two variables. As $w$ approaches 1, the formula quantifies the extent to which extreme values in one variable are associated with extreme values in the other. A larger $\lambda_U$ indicates stronger dependence in the upper tails.

Similarly, the lower tail dependence coefficient ($\lambda_L$) quantifies the degree of dependence between two variables when both are near the lower ends of their distributions, that is, when both variables take on very small values. This coefficient can be computed using the following formula:

$$\lambda_L = \lim_{w \to 0^+} \frac{C(w, w)}{w}$$

In this expression, $w$ again represents the probability level, but here it approaches 0, indicating that we are examining the behavior in the lower tails of the marginal distributions (i.e., $u, v \to 0$). The copula function $C(w, w)$ describes the joint behavior of the two variables as they both take on small values. A nonzero $\lambda_L$ suggests that there is a significant degree of dependence between the two variables in the lower tails of their distributions.

Thus, the upper and lower tail dependence coefficients provide a more detailed view of how two variables co-move in the extremes, beyond what can be captured by linear correlation measures.

### 3.4 OPTIMIZATION OBJECTIVE WITH COPULA ENTROPY

In `CoPruning`, we integrate Copula entropy, a concept introduced by Ma & Sun (2011), to account for the complexity and information content within the dependency structure. The second term $K_{\mathrm{II}} =$

$\alpha\|\mathcal{H}_c(\mathbf{s}, k_r) - \mathcal{H}_c(\mathbf{s}, \bar{k}_r)\|^2$ represents the difference in Copula entropy before and after pruning, where $\mathcal{H}_c$ denotes the Copula entropy. This term helps to maintain the structural complexity of the dependencies in the network.

The Copula entropy in (10) is expressed as

$$\mathcal{H}_c = -\int_{[0,1]^n} C(u_1, u_2, \ldots, u_n) \log C(u_1, u_2, \ldots, u_n) \, du_1 \, du_2 \, \ldots \, du_n \tag{15}$$

According to Ma & Sun (2011), Copula entropy quantifies the information contained in the dependence structure of random variables, making it a valuable measure in our optimization framework. By minimizing the difference in Copula entropy before and after pruning, we can ensure that the model retains its original dependency structure. This approach preserves the statistical properties of the network and maintains its performance.

## 4 ALGORITHM DESIGN

The proposed algorithm `CoPruning` tackles the network pruning problem as defined in (8). Inspired by Chen et al. (2022), `CoPruning` incrementally adjusts the sparsity of the weight vector $w$ by using a descending sequence of non-zero elements $r_0, \ldots, r_T$. It localizes weights and gradients to form joint distributions and copula entropy, allowing for a more precise evaluation of redundant parameters in neural networks. By computing the differences in these joint distributions and copula entropy at each pruning stage, `CoPruning` effectively reduces the network complexity while maintaining or enhancing performance. The following pseudocode outlines the steps of the `CoPruning` algorithm:

---
**Algorithm 1** Copula function based Pruning (CoPruning)

---
**Input:** Number of pruning stages $T$, initial weights $\bar{w}$, target sparsity $sp$, regularization parameters $\lambda$, $\epsilon$, batches $B_0, B_1, \ldots, B_T$, optimization step size $\tau > 0$
**Output:** Post-pruning weights $w$, satisfying $\|w\| \leq sp$
1: Set $r_0, r_1, \ldots, r_T$ as a descending sequence, with $r_0 > p$ and $r_T = sp$
2: **for** $t = 0$ to $T$ **do**
3:     Compute localized gradients $l_r = [\nabla\ell_1(\bar{k}_r), \ldots, \nabla\ell_n(\bar{k}_r)]^T$ using batch $B_t$
4:     Use local optimization approach $k_r^{(0)} \leftarrow w$, $\bar{k}_r \leftarrow \bar{w}$, localized gradients $s_r^{(0)} \leftarrow l_r$
5:     Construct joint distributions $C_i(\mathbf{s}, k_r^{(t)})$ and $C_i(\mathbf{s}, \bar{k}_r)$
6:     Compute Copula entropies $\mathcal{H}_c(\mathbf{s}, k_r^{(t)})$, $\mathcal{H}_c(\mathbf{s}, \bar{k}_r)$
7:     Compute the derivative of the pruning function with respect to $k_r$
8:     $\nabla Q \leftarrow 2\sum_{i=1}^{n}(C_i(\mathbf{s}, k_r^{(t)}) - C_i(\mathbf{s}, \bar{k}_r))\frac{\partial C_i}{\partial k_r} + 2\alpha(\mathcal{H}_c(\mathbf{s}, k_r^{(t)}) - \mathcal{H}_c(\mathbf{s}, \bar{k}_r))\frac{\partial \mathcal{H}_c}{\partial k_r} + 2\lambda(k_r^{(t)} - \bar{k})$
9:     $k_r^{(t+1/2)} \leftarrow k_r^{(t)} - \tau\nabla Q$
10:     $k_r^{(t+1)} \leftarrow$ Select from $k_r^{(t+1/2)}$ the $r_t$ components with the largest absolute values; set others to zero
11: **end for**
12: $w \leftarrow k_r^{(T+1)}$

---

**Weights Update and Step Size Selection.** The weights $k_r$ are updated using stochastic gradient descent (SGD) paired with the iterative hard thresholding (IHT) algorithm. For simplicity, we denote the derivative of $Q(k)$ as $\nabla Q$, with a detailed derivation available in A.3. The expression for $\nabla Q$ is given by:

$$\nabla Q = 2\sum_{i=1}^{n}(C_i(\mathbf{s}, k_r) - C_i(\mathbf{s}, \bar{k}_r))\frac{\partial C_i}{\partial k_r} + 2\alpha(\mathcal{H}_c(\mathbf{s}, k_r) - \mathcal{H}_c(\mathbf{s}, \bar{k}_r))\frac{\partial \mathcal{H}_c}{\partial k_r} + 2\lambda(k_r - k) \tag{16}$$

Following the weight updates driven by SGD (as seen in line 9 of Algorithm 1), the IHT method is applied. IHT retains the top $r_t$ components of the weight vector $k_r$ with the largest magnitudes and sets the remaining components to zero, ensuring adherence to the sparsity criteria. A crucial step in

the optimization process is the choice of the step size $\tau$ (referenced in line 9), as also suggested in Chen et al. (2022).

At the final stage of the algorithm, the localized weights $k_r$ are mapped back to the original weight vector $w$. This mapping ensures a one-to-one correspondence between the positions in $w$ and $k_r$, preserving the largest $sp$ elements in $k_r$ within $w$.

## 5 NUMERICAL RESULT

Table 1: Performance Comparison Across Different Sparsity Levels

| Network | Sparsity | MP | WF | CBS | CHITA | LR | CoPruning (ours) |
|---|---|---|---|---|---|---|---|
| MLPNet on MNIST (93.97%) | 0.5 | 93.93 | 94.02 | 93.96 | 93.97 | **95.26** ($\pm$ 0.03) | **95.26** ($\pm$ 0.05) |
| | 0.6 | 93.78 | 93.82 | 93.96 | 93.94 | **95.13** ($\pm$ 0.02) | **95.13** ($\pm$ 0.07) |
| | 0.7 | 93.62 | 93.77 | 93.98 | 93.80 | **94.93** ($\pm$ 0.03) | **94.98** ($\pm$ 0.06) |
| | 0.8 | 92.89 | 93.57 | 93.90 | 93.59 | **94.82** ($\pm$ 0.04) | **94.91** ($\pm$ 0.12) |
| | 0.9 | 90.30 | 91.69 | 93.14 | 92.46 | **94.32** ($\pm$ 0.05) | **94.34** ($\pm$ 0.15) |
| | 0.95 | 83.64 | 85.54 | 88.92 | 88.09 | 92.82 ($\pm$ 0.06) | **92.92** ($\pm$ 0.09) |
| | 0.98 | 32.25 | 38.26 | 55.45 | 46.25 | 84.43 ($\pm$ 0.10) | **86.34** ($\pm$ 0.20) |
| ResNet20 on CIFAR10 (91.36%) | 0.5 | 88.44 | 90.23 | 90.58 | 90.60 | **92.06** ($\pm$ 0.04) | **91.97** ($\pm$ 0.06) |
| | 0.6 | 85.24 | 87.96 | 88.88 | 89.22 | **91.98** ($\pm$ 0.09) | **91.95** ($\pm$ 0.10) |
| | 0.7 | 78.79 | 81.05 | 81.84 | 84.12 | 91.09 ($\pm$ 0.10) | **91.52** ($\pm$ 0.16) |
| | 0.8 | 54.01 | 62.63 | 51.28 | 57.90 | 89.00 ($\pm$ 0.12) | **90.38** ($\pm$ 0.19) |
| | 0.9 | 11.79 | 11.49 | 13.68 | 15.60 | **87.63** ($\pm$ 0.11) | 87.52 ($\pm$ 0.21) |
| | 0.95 | - | - | - | - | 80.25 ($\pm$ 0.17) | **81.60** ($\pm$ 0.14) |
| | 0.98 | - | - | - | - | 68.15 ($\pm$ 0.27) | **70.02** ($\pm$ 0.18) |
| ResNet50 on CIFAR10 (92.78%) | 0.95 | - | - | - | - | 83.75 ($\pm$ 0.14) | **84.37** ($\pm$ 0.11) |
| | 0.98 | - | - | - | - | 81.04 ($\pm$ 0.14) | **83.23** ($\pm$ 0.09) |
| MobileNet V1 on ImageNet (71.95%) | 0.5 | 62.61 | 68.91 | 70.21 | 70.42 | **70.12** ($\pm$ 0.13) | 69.54 ($\pm$ 0.11) |
| | 0.6 | 41.94 | 60.90 | 66.37 | 67.30 | **70.05** ($\pm$ 0.22) | 68.51 ($\pm$ 0.20) |
| | 0.7 | 6.78 | 29.36 | 55.11 | 59.40 | **68.15** ($\pm$ 0.17) | 67.78 ($\pm$ 0.13) |
| | 0.8 | 0.11 | 0.24 | 16.38 | 29.78 | **65.72** ($\pm$ 0.19) | **65.69** ($\pm$ 0.12) |
| | 0.9 | - | - | - | - | 47.65 ($\pm$ 0.15) | **58.08** ($\pm$ 0.28) |

In this section, we present the experimental results of various pruning methods across different sparsity levels on multiple neural network architectures. Specifically, the table compares MP (Han et al., 2015), WF (Singh & Alistarh, 2020), CBS (Yu et al., 2022), CHITA (Benbaki et al., 2023), LR (Chen et al., 2022), and our proposed method `CoPruning` on MLPNet (MNIST dataset), ResNet20 and ResNet50 (CIFAR-10 dataset), and MobileNet V1 (ImageNet dataset). The LR method is implemented in You & Cheng (2024), and its pruning data is included in the table for comparison. Best performance is indicated by boldface, allowing a margin of 0.1 for slight variations.

Based on the data in table 1, we observe that our proposed `CoPruning` method performs similarly to the Linear Regression (LR) method at low sparsity levels. However, at higher sparsity levels,

the performance of the `CoPruning` method is significantly better than that of the LR method. Particularly at sparsity levels of 0.9 and 0.95, `CoPruning` shows a notable increase in accuracy compared to LR, demonstrating the robustness and efficacy of the method under extreme sparsity conditions. Our experiments were conducted on a range of pre-trained neural network models, each chosen for their widespread use in benchmarking. The models include MLPNet (30K parameters) trained on the MNIST dataset (Lecun et al., 1998), ResNet20 (200K parameters) and ResNet50 (25M parameters) (He et al., 2016), both trained on the CIFAR-10 dataset (Krizhevsky, 2009), and MobileNetV1 (4.2M parameters) (Howard et al., 2017), trained on the ImageNet dataset (Deng et al., 2009). Each network was selected to represent a variety of architectures and scales, providing a comprehensive evaluation across different types of models and data. For more detailed experimental settings, please refer to A.4.

The results highlight the effectiveness of the `CoPruning` method, particularly in conditions of extreme sparsity. When sparsity levels reach 0.9 and above, `CoPruning` significantly outperforms traditional methods, maintaining high accuracy even in highly sparse networks. This robustness makes `CoPruning` particularly suitable for deployment in resource-constrained environments, where maintaining model accuracy despite pruning is critical.

Table 2: Performance at High Sparsity Levels with Noise

| Network | Sparsity | EWR | CoPruning (proposed) |
|---------|----------|-----|----------------------|
| MLPNet (93.97%) | $0.95 + \sigma$ | 90.50 ($\pm$ 0.07) | **92.82** ($\pm$ 0.09) |
| | $0.98 + \sigma$ | 83.69 ($\pm$ 0.10) | **85.94** ($\pm$ 0.16) |
| ResNet20 (91.36%) | $0.95 + \sigma$ | 79.05 ($\pm$ 0.16) | **81.49** ($\pm$ 0.22) |
| | $0.98 + \sigma$ | 68.01 ($\pm$ 0.25) | **70.26** ($\pm$ 0.31) |
| ResNet50 (92.78%) | $0.95 + \sigma$ | 84.92($\pm$ 0.22) | **85.78** ($\pm$ 0.22) |
| | $0.98 + \sigma$ | 82.94 ($\pm$ 0.17) | **85.59** ($\pm$ 0.19) |
| MobileNet (71.95%) | $0.8 + \sigma$ | 63.62 ($\pm$ 0.15) | **65.59** ($\pm$ 0.19) |
| | $0.9 + \sigma$ | 47.98 ($\pm$ 0.16) | **58.26** ($\pm$ 0.23) |

The results in Table 2 illustrate the performance of EWR (You & Cheng, 2024) and our proposed method at high sparsity levels with noise. In particular, noise $\sigma$ (representing 20% of parameters with noise) is added at sparsity levels 0.95 and 0.98 across four different network architectures: MLPNet, ResNet20, ResNet50, and MobileNet. It can be observed that `CoPruning` consistently outperforms EWR in most cases under high sparsity conditions, even with added noise.

Table 3: Accuracy Comparison of ResNet20 with and without Local optimization

| Local optimization Parameter $r$ | No Local Opt. (%) | With Local Opt. (%) |
|----------------------------------|-------------------|---------------------|
| 10 | | 60.79 |
| 50 | | 63.84 |
| 100 | | 66.33 |
| 200 | 69.58 | 68.49 |
| 500 | | 70.32 |
| 1000 | | 70.71 |
| 1500 | | 70.02 |
| $\sqrt{n}$ | | **70.72** |

The results in Table 3 show the accuracy comparison of ResNet20 with and without the Local optimization strategy, under a target sparsity of 0.98. The Local optimization parameter $r$ represents the size of the local area, and as $r$ increases, the effect of the Local optimization strategy becomes more extensive. The data in the table reflect the performance of ResNet20 under different Local optimization parameters. Without Local optimization, the network's accuracy does not change, while with the Local optimization strategy applied, the network's accuracy significantly improves as the Local optimization parameter $r$ increases. For example, at $r = 10$, the accuracy reaches 60.79%, and at $r = n$ (representing globalization), the accuracy reaches the highest value of 70.72%. This indicates that the Local optimization strategy effectively improves the network's performance under noise and high sparsity with selecting an appropriate Local optimization parameter.

## 6 CONCLUSIONS AND FUTURE IMPACT

In this paper, we introduced the `CoPruning` method, which exhibits significant advantages in high-sparsity settings and performs well even in the presence of noise. Our experiments demonstrated that `CoPruning` maintains high accuracy at extreme sparsity levels, outperforming traditional methods like Linear Regression (LR) and Entropic Wasserstein Regression (EWR). It proved effective across various neural network architectures, showcasing its robustness and suitability for challenging pruning conditions.

Looking forward, `CoPruning`'s ability to handle high sparsity and noise makes it a promising tool for deploying neural networks in resource-limited environments where memory and computation are critical. The principles behind `CoPruning` may inspire further research in optimizing neural networks for edge computing and other scenarios requiring efficient compression and robustness. Future work could explore its application to more diverse tasks and datasets, as well as its integration with other compression and optimization techniques to expand its potential.

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

# A  APPENDIX

## A.1  NONLINEAR PROOF OF THE RELATIONSHIP BETWEEN PARAMETERS AND GRADIENTS

The nonlinear relationship between model parameters and gradients in neural networks can be explained through several mathematical components. Starting with the neural network's output, which is typically given by:

$$y = f(Wx + b)$$

where $W$ is the weight matrix, $x$ is the input vector, $b$ is the bias term, and $f$ is the activation function. The most common activation functions, such as ReLU, Sigmoid, and Tanh, are nonlinear. The loss function, denoted as $L$, is a measure of how far the predicted output $y$ is from the target $t$. For simplicity, using a quadratic loss function:

$$L = \frac{1}{2}(y - t)^2$$

we calculate the gradient of $L$ with respect to $W$ through the chain rule:

$$\frac{\partial L}{\partial W} = (y - t) \cdot f'(Wx + b) \cdot x$$

Here, $f'(Wx+b)$ is the derivative of the activation function, and since $f$ is nonlinear, $f'(Wx+b)$ is also nonlinear. This introduces a nonlinear relationship between the parameters $W$ and the gradient $\frac{\partial L}{\partial W}$, implying that the gradient does not change linearly with $W$.

In deeper networks, backpropagation involves the propagation of the gradients through multiple layers. For each layer $l$, the gradient at a hidden layer depends on the chain rule applied to all subsequent layers:

$$\frac{\partial L}{\partial W_l} = \frac{\partial L}{\partial h_l} \cdot f'(h_l) \cdot x_l$$

where $h_l$ is the input to the $l$-th layer, and $f'(h_l)$ is the derivative of the activation function at that layer. Because $f'(h_l)$ is nonlinear, each layer introduces additional nonlinearity into the gradients, making the relationship between the final parameters and gradients even more complex and nonlinear.

The nonlinearity is further exacerbated by the choice of the loss function. For instance, a widely used loss function in classification tasks is cross-entropy:

$$L = -\sum t \log(y)$$

The gradient of this loss function with respect to $W$ is:

$$\frac{\partial L}{\partial W} = -\sum \frac{t}{y} \cdot \frac{\partial y}{\partial W}$$

Since $y$ is a nonlinear function of $W$, the gradient of the cross-entropy loss also exhibits nonlinearity with respect to the model parameters.

Modern optimization algorithms, such as Adam and RMSprop, introduce further complexity in the parameter update rule. Adam, for example, updates weights using the following equation:

$$W_{t+1} = W_t - \alpha \frac{\hat{m}_t}{\sqrt{\hat{v}_t} + \epsilon}$$

where $\hat{m}_t$ and $\hat{v}_t$ are the first and second moment estimates of the gradients, respectively, and $\alpha$ is the learning rate. These moment-based adjustments, which rely on previous gradients, introduce additional nonlinearity into the parameter update process. Therefore, even though Adam uses gradients for optimization, the relationship between the parameter updates and gradients becomes highly nonlinear due to these adaptive adjustments.

In summary, the combination of nonlinear activation functions, the accumulation of nonlinearities through backpropagation, nonlinear loss functions, and adaptive optimization algorithms collectively result in a highly nonlinear relationship between the parameters and gradients in neural networks.

## A.2 PROOF OF SKLAR'S THEOREM

**Proposition 1 (Sklar's Theorem)** Sklar's Theorem (Sklar, 1959) states that any multivariate joint distribution can be decomposed into its marginal distributions and a Copula function. Specifically, let $H(x_1, x_2, \ldots, x_n)$ be an $n$-dimensional joint distribution function, and let $F_1(x_1), F_2(x_2), \ldots, F_n(x_n)$ be its marginal distribution functions. Then there exists a Copula function $C(u_1, u_2, \ldots, u_n)$ such that:

$$H(x_1, x_2, \ldots, x_n) = C(F_1(x_1), F_2(x_2), \ldots, F_n(x_n)) \tag{17}$$

Here, the Copula function $C$ captures the dependency structure between the marginal distributions. This property makes Copula functions particularly valuable in our framework, where preserving the dependency structure between network parameters is essential for maintaining performance during pruning.

Sklar's Theorem establishes a fundamental relationship between any multivariate joint distribution function and its marginals through a Copula function. First, let's define the pseudo-inverse $F_i^{-1}$ of each marginal distribution function $F_i$ by

$$F_i^{-1}(u_i) = \inf\{x_i \in \mathbb{R} : F_i(x_i) \geq u_i\}$$

for $u_i \in [0, 1]$. We construct a Copula function $C : [0, 1]^n \to [0, 1]$ using these inverses and the joint distribution function $H$, such that

$$C(u_1, u_2, \ldots, u_n) = H(F_1^{-1}(u_1), F_2^{-1}(u_2), \ldots, F_n^{-1}(u_n)).$$

To validate that $C$ is indeed a Copula, we need to demonstrate it is both grounded and $n$-increasing. The function $C$ is grounded because if any $u_i = 0$, then $C(u_1, \ldots, u_n) = 0$. It is $n$-increasing as the volume $V_C$ under the Copula over any hyperrectangle in $[0, 1]^n$ is non-negative, which follows from the $n$-increasing nature of $H$ and the non-decreasing property of $F_i^{-1}$. The uniform margins condition is satisfied because the projection of $C$ over any axis returns $u_i$, which means

$$C(1, \ldots, 1, u_i, 1, \ldots, 1) = u_i.$$

Finally, to prove that the joint distribution $H$ can be completely reconstructed using $C$ and the marginal distributions $F_i$, we observe that substituting $F_i(x_i)$ into $C$ gives

$$C(F_1(x_1), F_2(x_2), \ldots, F_n(x_n)) = H(F_1^{-1}(F_1(x_1)), F_2^{-1}(F_2(x_2)), \ldots, F_n^{-1}(F_n(x_n))).$$

Since $F_i^{-1}(F_i(x_i)) = x_i$ almost everywhere, particularly when $F_i$ are continuous, we obtain

$$H(x_1, x_2, \ldots, x_n) = C(F_1(x_1), F_2(x_2), \ldots, F_n(x_n)),$$

thus completing the proof of Sklar's Theorem. This result shows that the Copula $C$ effectively captures all dependencies between the variables as encoded by $H$.

## A.3  DERIVATIVE OF $Q(k)$

For the weight matrix $W$ and the gradient matrix $L$, a given parameter $r$ is used to compute local sums and normalize each element. The computation follows these formulas:

For the weight matrix $W$:

$$\text{Sum}_{i,j} = \sum_{m=-\lfloor r/2 \rfloor}^{\lfloor r/2 \rfloor} \sum_{n=-\lfloor r/2 \rfloor}^{\lfloor r/2 \rfloor} |W_{i+m,j+n}| \tag{18}$$

$$k_{i,j} = \frac{|W_{i,j}|}{\text{Sum}_{i,j}} \tag{19}$$

For the gradient matrix $L$:

$$\text{Sum}_{i,j}^{L} = \sum_{m=-\lfloor r/2 \rfloor}^{\lfloor r/2 \rfloor} \sum_{n=-\lfloor r/2 \rfloor}^{\lfloor r/2 \rfloor} |L_{i+m,j+n}| \tag{20}$$

$$s_{i,j} = \frac{|L_{i,j}|}{\text{Sum}_{i,j}^{L}} \tag{21}$$

These formulas allow the transformation of the original matrices into their normalized forms $k$ and $s$, using localized sums based on the neighborhood size specified by $r$.

The optimization problem $Q(k)$ leverages the normalized matrices $s$ and $k$ obtained from the previous calculations. The objective function and its constraints are defined as:

$$\min_{k} Q(k) = \sum_{i=1}^{n} \|C_i(s_r, k_r) - C_i(s_r, \bar{k}_r)\|^2 + \alpha \|\mathcal{H}_c(s_r, k_r) - \mathcal{H}_c(s_r, \bar{k}_r)\|^2 + \lambda \|k_r - \bar{k}_r\|^2 \tag{22}$$

The derivative of $Q(k)$ with respect to $k_r$ is calculated as follows:

$$\nabla Q = 2 \sum_{i=1}^{n} (C_i(s_r, k_r) - C_i(s_r, \bar{k}_r)) \frac{\partial C_i}{\partial k_r} + 2\alpha (\mathcal{H}_c(s_r, k_r) - \mathcal{H}_c(s_r, \bar{k}_r)) \frac{\partial \mathcal{H}_c}{\partial k_r} + 2\lambda (k_r - k) \tag{23}$$

This formulation provides the necessary framework to compute the gradient of the objective function, which is essential for optimizing $k_r$ effectively.

The Frank Copula for variables $s_r$ and $k_r$ is given by:

$$C(s_r, k_r) = -\frac{1}{k_r} \log \left( 1 + \frac{(e^{-k_r s_r} - 1)(e^{-k_r(1-s_r)} - 1)}{e^{-k_r} - 1} \right) \tag{24}$$

To simplify the derivative calculation, we define three intermediate functions:

$$f(k_r) = e^{-k_r s_r} - 1, \quad g(k_r) = e^{-k_r(1-s_r)} - 1, \quad h(k_r) = e^{-k_r} - 1$$

Using these definitions, the derivative of the Frank Copula with respect to $k_r$ is calculated as follows:

$$\frac{\partial C}{\partial k_r} = \frac{1}{k_r^2} \log \left( 1 + \frac{f(k_r)g(k_r)}{h(k_r)} \right)$$
$$- \frac{1}{k_r} \left( \frac{(f'(k_r)g(k_r) + f(k_r)g'(k_r)) h(k_r) - f(k_r)g(k_r)h'(k_r)}{h(k_r)^2} \right)$$

Where the derivatives of $f$, $g$, and $h$ are:

$$f'(k_r) = -s_r e^{-k_r s_r}, \quad g'(k_r) = -(1 - s_r)e^{-k_r(1-s_r)}, \quad h'(k_r) = -e^{-k_r}$$

This formulation helps clarify the derivative calculation by isolating the exponential terms and their interactions.

The entropy $\mathcal{H}_c$ of a copula $C$ can generally be expressed as:

$$\mathcal{H}_c = -\int\int C(u, v; k_r) \log C(u, v; k_r) \, du \, dv$$

This formula represents the entropy measure of the dependence structure encoded by the copula $C$, where $u$ and $v$ are the marginal distributions integrated over their respective domains.

The derivative of the copula entropy with respect to the parameter $k_r$ involves calculating the rate of change of the entropy as the copula parameter changes. It is given by:

$$\frac{\partial \mathcal{H}_c}{\partial k_r} = -\int\int \left( \frac{\partial C(u, v; k_r)}{\partial k_r} \log C(u, v; k_r) + \frac{\frac{\partial C(u,v;k_r)}{\partial k_r}}{C(u, v; k_r)} \right) du \, dv$$

This derivative takes into account both the direct effect of changes in $k_r$ on $C$ and the change in the entropy due to the adjustment of the copula function.

## A.4 EXPERIMENTAL SETUP

The experimental framework leverages robust hardware configurations to manage the demanding computation required for model training and pruning. Initially, the models—MLPNet and ResNet20—are trained using a single NVIDIA RTX A4080 16 GB GPU, while ResNet50 and MobileNetV1 are trained on an NVIDIA RTX A5000 24 GB GPU. The training durations were approximately 0.5 hours for both MLPNet and ResNet20, while ResNet50 and MobileNetV1 required around 1 day each, highlighting the significant computational effort, especially when handling ImageNet data. The pre-pruning accuracy for each model is systematically documented in Table 1.

For the pruning phases, we utilized a single NVIDIA RTX A4080 16 GB GPU. Given the intensive nature of training and pruning MobileNetV1, employing efficient processing strategies is strongly advised to optimize resource utilization and efficiency. Our pruning method resulted in a 4x reduction in memory usage compared to the original model.

In the detailed pruning schedule outlined in Table 1, we specified the pruning stages for LR and `CoPruning` to be 15 for MLPNet and ResNet20, 10 for MobileNetV1, and 10 for ResNet50. In Table 2, the pruning stages for `CoPruning` is set to 15 for ResNet50, and remain the same as described above for other models. The sparsity levels $k_1, k_2, \ldots, k_T$ in Algorithm 1 adhere to an exponential gradual pruning schedule $k_t = k_T + (k_0 - k_T) \cdot (1 - \frac{t}{T})^3$, with the initial sparsity $k_0$ set to zero. Additionally, the fisher sample size is configured as per the suggestions in You & Cheng (2024), which is replicated in Table 4 of this document for reference.

Table 4: Model Configurations for Various Methods

| Model | WF & CBS | LR & EWR | CoPruning |
|---|---|---|---|
| | Sample / Batch | Sample / Batch | Sample / Batch |
| MLPNet | 1000 / 1 | 1000 / 1 | 1000 / 1 |
| ResNet20/50 | 1000 / 1 | 1000 / 1 | 1000 / 1 |
| MobileNet | 400 / 2400 | 1000 / 16 | 1000 / 16 |

## A.5 VISUALIZING PARAMETER DISTRIBUTIONS USING COPULA DURING PRUNING PHASES

In the pruning process of the ResNet20 network trained on CIFAR-10, we utilized Copula functions to model and preserve dependencies among the network parameters at various pruning stages. This section visually presents the effectiveness of Copula functions in fitting the parameter distributions at stages 1, 4, 8, and 12.

In addition to capturing the interdependencies, we also examine the marginal distributions of $s$ and $k$, which represent localized transformations of the parameters $w$ and $l$, respectively. These marginal distributions allow us to better understand the behavior of individual parameters during the pruning process. By analyzing $s$ and $k$, we can gain insights into how individual parameter distributions evolve and how pruning affects localized regions of the network.

The following figures demonstrate the Copula-based joint parameter distributions as well as the marginal distributions of $s$ and $k$ at key pruning stages, providing a comprehensive view of how parameter dependencies and localized behaviors are preserved throughout the process.

Figure 2 shows the Copula-based joint parameter distributions at stages 1, 4, 8, and 12, while Figure 3 displays the marginal distributions of $s$ and $k$, representing the localized parameter distributions.

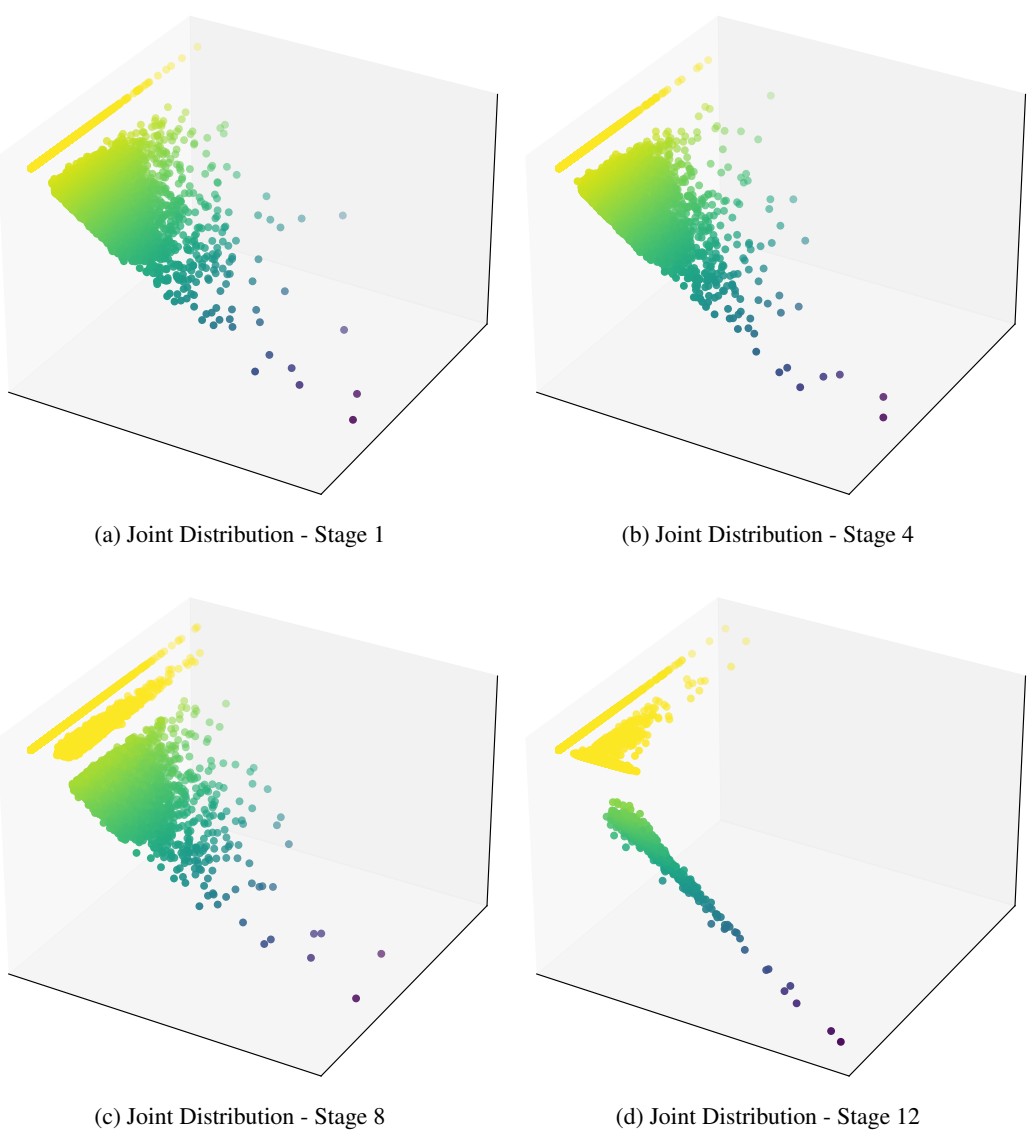

(a) Joint Distribution - Stage 1

(b) Joint Distribution - Stage 4

(c) Joint Distribution - Stage 8

(d) Joint Distribution - Stage 12

Figure 2: Copula-based joint parameter distributions during stages 1, 4, 8, and 12.

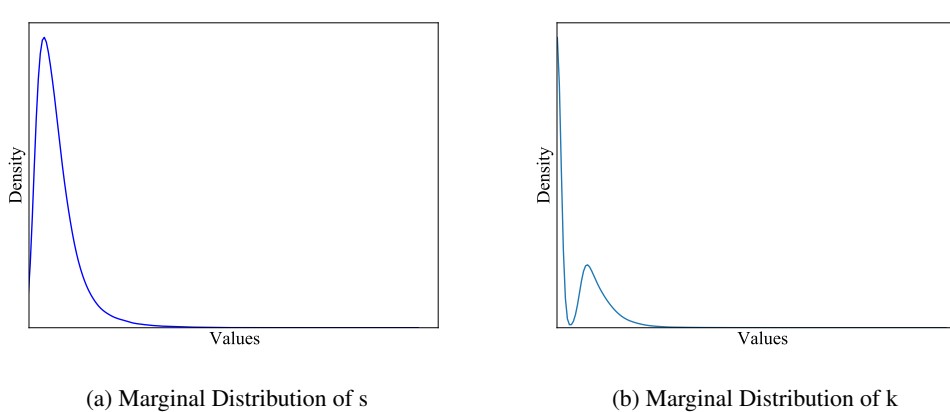

(a) Marginal Distribution of s                    (b) Marginal Distribution of k

Figure 3: Marginal distributions of 's' and 'k', representing the localized parameter distributions.

## A.6 COMPARISON OF PRUNING METHODS UNDER DIFFERENT SPARSITY LEVELS

This appendix section provides a detailed comparison of the CoPruning and EWR pruning methods applied to MLPNet and ResNet20 models under two different sparsity conditions, 25% and 40%. The accuracy is shown with standard deviation to indicate the variability of the results.

Table 5: Performance Comparison of CoPruning and EWR Pruning Methods under Different Sparsity Levels and Noise Conditions

| Model | Pruning Method | Accuracy (%) | | | |
|---|---|---|---|---|---|
| | | Sparsity 0.95 (30% Noise) | Sparsity 0.95 (50% Noise) | Sparsity 0.98 (30% Noise) | Sparsity 0.98 (50% Noise) |
| MLPNet | CoPruning | 92.87 ($\pm$ 0.14) | 92.93 ($\pm$ 0.11) | **85.72** ($\pm$ 0.20) | **85.81** ($\pm$ 0.19) |
| | EWR | 92.88 ($\pm$ 0.09) | 92.89 ($\pm$ 0.10) | 85.52 ($\pm$ 0.11) | 85.70 ($\pm$ 0.13) |
| ResNet20 | CoPruning | 81.22 ($\pm$ 0.24) | 81.28 ($\pm$ 0.15) | **69.23** ($\pm$ 0.18) | **69.42** ($\pm$ 0.21) |
| | EWR | 81.13 ($\pm$ 0.15) | 81.30 ($\pm$ 0.16) | 68.97 ($\pm$ 0.18) | 68.76 ($\pm$ 0.19) |

The slight increase in accuracy with higher noise levels is due to the role of noise in preventing overfitting and enhancing generalization, as noted by Neelakantan et al. (2015). Additionally, noise helps smooth gradient updates, allowing models to escape local minima and improve robustness during training (Zhang et al., 2023).

## A.7 AVERAGE LOSS COMPARISON

In this part, we provide a detailed comparison of the average loss across different noise configurations during the training process. We investigate how varying noise standard deviations and proportions affect the training dynamics under sparsity levels of 0.95 and 0.98.

Figure 4 illustrates the comparison of average loss across epochs for different sparsity levels, comparing noise standard deviations 6 (proportion 0.5) and 4 (proportion 0.25) for sparsity levels 0.95 and 0.98 respectively. Results are based on ResNet20 training on the CIFAR-10 dataset.

Figure 5 presents a similar comparison for MLPNet training on the MNIST dataset, highlighting the effect of noise configurations under the same sparsity levels.

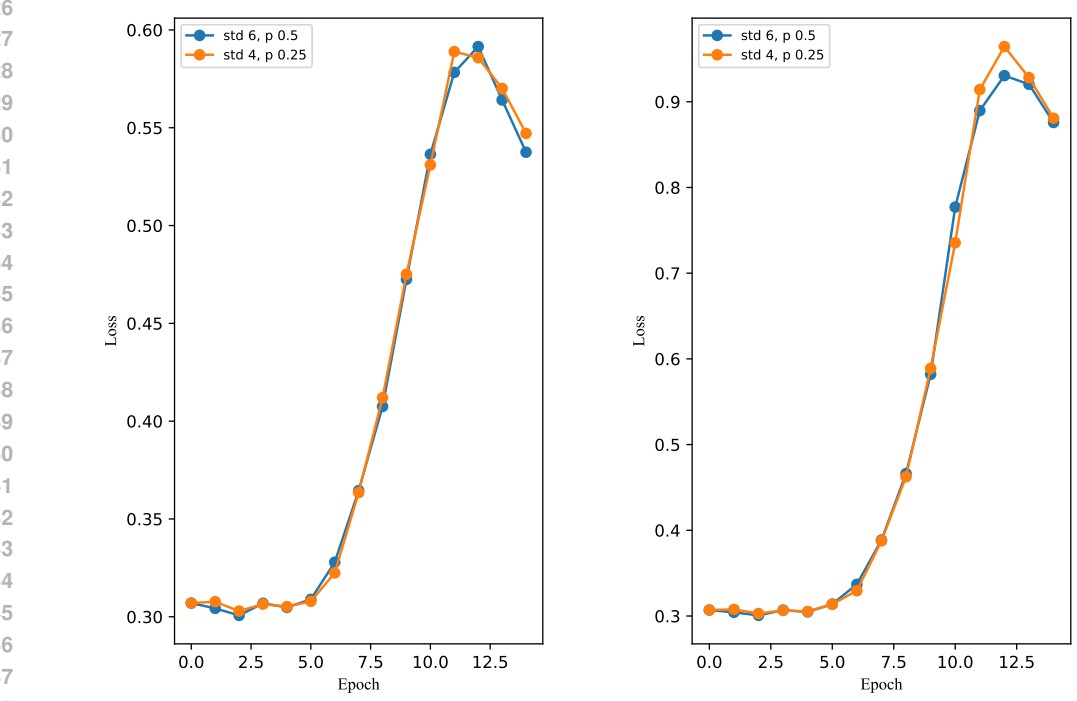

Figure 4: Comparison of average loss across epochs for different sparsity levels, comparing noise standard deviations 6 (proportion 0.5) and 4 (proportion 0.25) for sparsity levels 0.95 and 0.98 respectively. Results are based on ResNet20 training on the CIFAR-10 dataset.

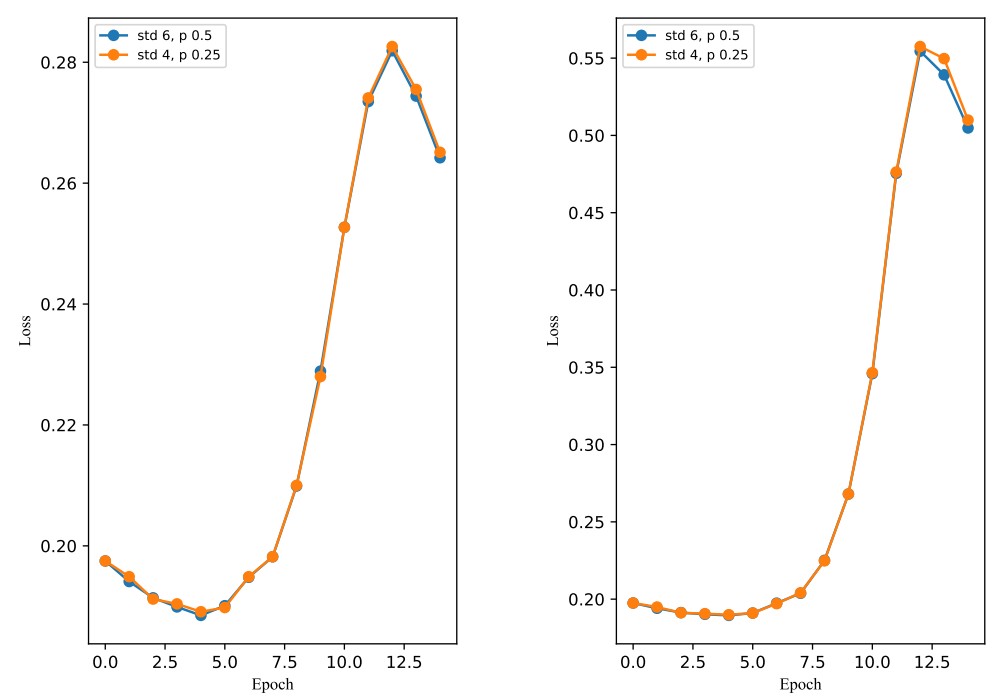

Figure 5: Comparison of average loss across epochs for different sparsity levels, comparing noise standard deviations 6 (proportion 0.5) and 4 (proportion 0.25) for sparsity levels 0.95 and 0.98 respectively. Results are based on MLPNet training on the MNIST dataset.

## A.8 Pruning Results for Different Models

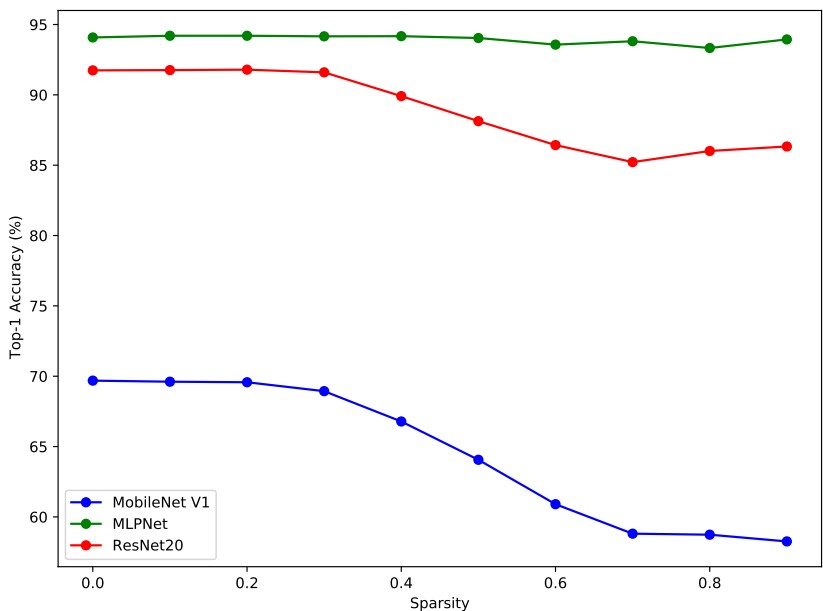

Figure 6: Top1 Accuracy vs Sparsity for Three Models

Similarly, Figure 7 shows the Top5 accuracy for the same models. The trend mirrors that of Top1 accuracy, with all models experiencing a decrease in accuracy as the sparsity increases. However, the Top5 accuracies tend to decline more gradually, maintaining better performance at higher sparsity levels.

In this appendix, we present the pruning results for three different models using the `CoPruning` method. The models included in this comparison are MobileNet v1, MLPNet on MNIST, and ResNet20 on CIFAR10.

The pruning process was designed to progressively reduce the number of parameters in each model to achieve a target sparsity level of 0.9. This was done over 9 pruning stages, where at each stage, a fraction of weights was pruned. At each sparsity level, the models were retrained, and the Top1 and Top5 accuracies were recorded. The results are shown in Figures 6 and 7.

Figure 6 shows the Top1 accuracy of the three models as a function of sparsity. As expected, as the sparsity increases, the accuracy of each model decreases, though the rate of decline varies between models. The target sparsity of 0.9 was achieved at the final stage.

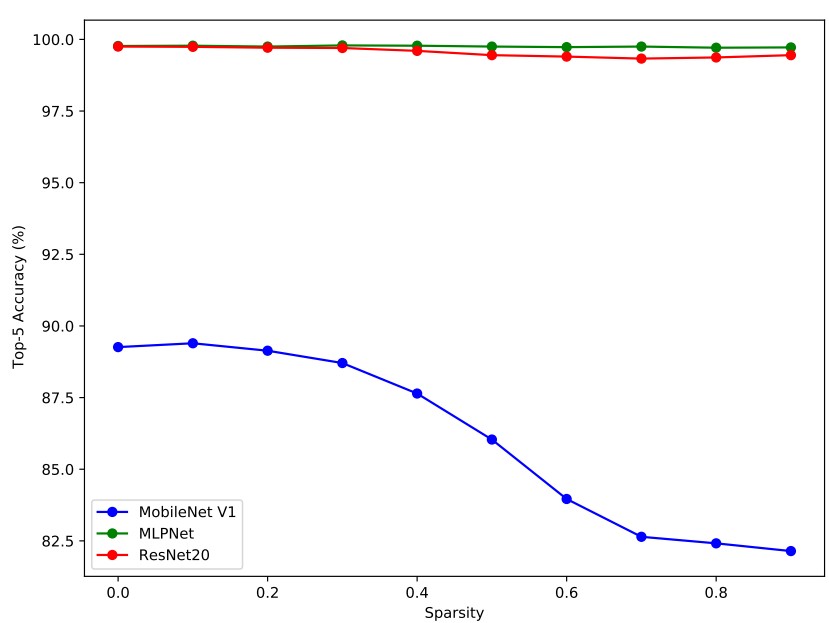

Figure 7: Top-5 Accuracy vs Sparsity for Three Models

