# OpenReview forum: "CoPruning: Exploring the Parameter-Gradient Nonlinear Correlation for Neural Network Pruning Using Copula Function"
_ICLR.cc/2025/Conference — ICLR 2025 Conference Withdrawn Submission_

### Official Review · Reviewer_z8Ee · 2024-11-01

**Soundness:** 3
**Presentation:** 2
**Contribution:** 2
**Rating:** 3
**Confidence:** 3

**Summary:**

This paper proposes a pruning framework using opula function to model nonlinear correlations between parameters and gradients which integrates local optimization to capture relative parameter changes within their local context. The proposed method combines copula-based joint distribution modeling and incorporates copula entropy to maintain structural complexity.

**Strengths:**

1. The algorithm pipeline is clear and understandable.
2. The theory analysis is solid and comprehensive.
3. The proposed method achieve a better performance at high sparsity levels and maintains accuracy in noisy conditions.

**Weaknesses:**

1. Tested neural network architectures are limited.
2. The proposed method seems works only at high sparsity in large dataset.
3. Lack of the discussion about the computational cost.

**Questions:**

1. Can proposed method still works well on attention based model (e.g., ViTs)?
2. Any explaination about why in lower sparsity the proposed method cannot outperform other methods (e.g., imagenet; sparsitiy 0.7,0.8)?
3. Compared with other methods, how is the computational cost between the proposed method with others (e.g., training time, gpu memory usage)?

---

### Official Review · Reviewer_ocPa · 2024-11-03

**Soundness:** 2
**Presentation:** 1
**Contribution:** 2
**Rating:** 3
**Confidence:** 3

**Summary:**

This paper introduces a novel pruning framework that leverages a copula function to capture nonlinear correlations between model parameters and gradients. This approach demonstrates notable efficacy, particularly at extreme pruning ratios; for instance, it achieves a substantial accuracy improvement of 10.43% at 0.9 sparsity.

**Strengths:**

* This work proposes a novel method to capture the correlation between parameters and gradients
* Their method is validated across different CNN architectures and datasets, achieving promising results, especially in extreme sparsity ratios scenarios.

**Weaknesses:**

* The notation usage is unclear. Specifically, the symbol $i$ is used inconsistently, serving as an index for weights and dataset samples in the related work section.
* There are typos in the algorithm. In line 4, it appears that $k_r^{(t)}$ should be used instead, and it would improve readability to move $\bar{k_r} \leftarrow \bar{\omega}$ outside the for-loop.
* The copula function lacks explanation, making it challenging to understand. In general, Section 3 requires more clarity and readability improvements.

**Questions:**

* In Lines 54–65, three factors are mentioned as influencing the nonlinearity between model parameters and gradients. Could you clarify the difference between the first two factors?
* In Line 130, $X$ is referred to as the gradient matrix. How does $X$ differ from $\nabla l_i$ from Lines 139–140?
* What does a "noisy environment" mean in the context of pruning? Isn’t the model always pruned from the same pre-trained model?
* For localized gradient results $s$, what is the definition of "localization"?
* In Line 1 of the Algorithm, could you clarify the meaning of $p$?

---

### Official Review · Reviewer_qkwG · 2024-11-03

**Soundness:** 3
**Presentation:** 3
**Contribution:** 3
**Rating:** 5
**Confidence:** 3

**Summary:**

The paper proposes a new framework for sparsity-aware training using copula function to capture relationship between the network parameters and its gradients, along with local optimization approach resulting in new metrics for optimization.

**Strengths:**

The proposed framework achieves significant improvement over SoTA pruning frameworks at high sparsity rates

**Weaknesses:**

While the paper shows substantial improvements over state-of-the-art methods at high sparsity rates, the overall degradation compared to the baseline network is so significant that the practical benefits of these improvements may be minimal. For instance, ResNet-20 on CIFAR-10 demonstrates a 1.87% improvement over the LR method at a sparsity level of 0.98. However, this still reflects a degradation of over 20% compared to the baseline, making the 1.87% gain practically negligible. Similarly, for MV1 on ImageNet, there is an improvement of over 10% compared to the LR method at 0.9 sparsity, but the degradation relative to the baseline remains over 13%.

**Questions:**

Given that MV1 on ImageNet shows a degradation of over 13% compared to the baseline, how would it perform if a smaller variant of MV1 (such as using a 0.75 depth multiplier) were used instead and pruned to achieve 0.6-0.8 sparsity? This approach might yield less degradation based on the results observed for both the proposed method and the LR method at similar sparsity levels?

Are there specific scenarios where CoPruning fails or performs suboptimally compared to simpler or less computationally intense methods?

---

### Note · Authors · 2024-11-13

I have read and agree with the venue's withdrawal policy on behalf of myself and my co-authors.